# Nitrergic Enteric Neurons in Health and Disease—Focus on Animal Models

**DOI:** 10.3390/ijms20082003

**Published:** 2019-04-24

**Authors:** Nikolett Bódi, Zita Szalai, Mária Bagyánszki

**Affiliations:** Department of Physiology, Anatomy and Neuroscience, Faculty of Science and Informatics, University of Szeged, H-6726 Szeged, Hungary; zszalai@bio.u-szeged.hu (Z.S.); bmarcsi@bio.u-szeged.hu (M.B.)

**Keywords:** nitric oxide synthase, nitrergic enteric neurons, enteric plexuses, microbiota–gut–brain axis, type 1 diabetes, alcoholism, gastrointestinal inflammation

## Abstract

Nitrergic enteric neurons are key players of the descending inhibitory reflex of intestinal peristalsis, therefore loss or damage of these neurons can contribute to developing gastrointestinal motility disturbances suffered by patients worldwide. There is accumulating evidence that the vulnerability of nitrergic enteric neurons to neuropathy is strictly region-specific and that the two main enteric plexuses display different nitrergic neuronal damage. Alterations both in the proportion of the nitrergic subpopulation and in the total number of enteric neurons suggest that modification of the neurochemical character or neuronal death occurs in the investigated gut segments. This review aims to summarize the gastrointestinal region and/or plexus-dependent pathological changes in the number of nitric oxide synthase (NOS)-containing neurons, the NO release and the cellular and subcellular expression of different NOS isoforms. Additionally, some of the underlying mechanisms associated with the nitrergic pathway in the background of different diseases, e.g., type 1 diabetes, chronic alcoholism, intestinal inflammation or ischaemia, will be discussed.

## 1. Introduction

In the gastrointestinal (GI) tract, as in most other organ systems, nitric oxide (NO) plays a defining role in regulating several functions in both physiological and pathological states. NO contributes to the maintenance of GI mucosal integrity and circulation, regulation of secretion, smooth muscle function or mucosal inflammation [1,2]. NO, produced by neuronal NO synthase (nNOS), is one of the main inhibitory neurotransmitters in GI smooth muscle [3,4].

The involvement of nNOS and NO pathways in enteric neuropathies was well-described in a variety of disorders, like esophageal or internal anal sphincter achalasia, hypertrophic pyloric stenosis, gastroparesis, Chagas’, or Hirschsprung’s disease [5]. This review will provide information about the role of enteric neuronal NO in health and different pathological animal models with type 1 diabetes, chronic alcoholism, intestinal inflammation, or ischaemia.

## 2. Enteric Neurons

Enteric neurons reside in two ganglionated plexuses of the enteric nervous system (ENS). The myenteric plexus is situated between the longitudinal and circular muscle layers of the gut tube and regulates the motility of these smooth muscles. The submucous plexus is embedded in the GI submucosal layer and coordinates the absorption, secretion and circulation of the gut wall [6]. The enteric ganglia (Figure 1) contains enteric glia cells and neurons, connected by interganglionic segments. The number of enteric neurons is equal to the number of spinal cord neurons in the same species [7].

Intrinsic primary sensory neurons, interneurons and motoneurons are all present in the ENS, therefore the enteric neurons are able to form local reflex circuits in the intestinal wall and can work autonomously from the central nervous system [6,8]. At the same time, the ENS bidirectionally communicates with the central nervous system [9,10], and several data support the intimate connection of ENS and gut microbiota, therefore these three systems make a microbiota–gut–brain axis [11,12].

The neurochemical phenotype of enteric neurons is critical to their function [13,14]. The neurochemical composition of the ENS is much more diverse than the sympathetic and parasympathetic divisions of the autonomic nervous system. From this point of view, the ENS is similar to the central nervous system, since the whole range of the classical and other neurotransmitters are present in enteric neurons [13,15].

### Nitrergic Enteric Neurons

NO can be synthesized by nNOS, endothelial NOS (eNOS) and inducible NOS (iNOS) in the different cell types in the body. In the ENS, the major inhibitory non-adrenergic, non-cholinergic (NANC) neurotransmitter is endogenous NO [4,16,17,18]. In enteric neurons, nNOS is the predominant form that produces NO in a physiological state [14,19,20,21,22], but all three isoforms of the NOS are present in mRNA and protein levels (Figure 2) in the enteric neurons. Different pathological states might affect the role of the NOS isoforms in the ENS [23].

The proportion of the nitrergic subpopulation when compared to the total neuronal cell number is shown to be different in the two enteric plexuses. The nitrergic neurons account for only a few percent of all neurons in the submucous plexus [24], while the proportion of the nitrergic myenteric neurons (Figure 3) is significant (23–52%). This massive proportion of the nitrergic myenteric neurons can be explained by the function of this neuron population. Most of the nitrergic myenteric neurons are inhibitory interneurons or inhibitory motoneurons innervating the muscle layers of the alimentary tract [14,25].

## 3. Pathophysiology of Nitrergic Enteric Neurons

### 3.1. Type 1 Diabetes

It is well established that NOS-containing myenteric neurons are curiously susceptible to diabetic injuries [26,27,28] and impaired nitrergic innervation is accompanied by motility dysfunction [29]. Moreover, the nitrergic neurons located in different gut segments display strictly region-specific responsiveness to the diabetic state and also to immediate insulin replacement [27]. In the jejunum, ileum and colon of diabetic rats, both the nitrergic and total number of myenteric neurons were decreased assuming diabetes-related cell loss in these segments. However, in the duodenum of diabetic rats, the decreased number of nitrergic neurons was not accompanied by changes in the total number of myenteric neurons, presuming region-specific neurochemical modification of neurons here [27] (Table 1). Several other studies confirm diabetes-associated decreases of nitrergic myenteric neurons. nNOS neurons are reduced in the antrum and jejunum in spontaneously diabetic Bio-breeding rats [28], in the ileum of diabetic dogs [30], and in the human appendix [31], and reduced nNOS protein and mRNA expression were observed in the stomach of mice [29,32] withP2×7 receptor-mediated diabetic damage of nitrergic neurons [32]. Besides nitrergic neuronal damage, the diabetic animals represented delayed gastric emptying [29] and faster small intestinal and colonic transit compared to controls [27,33].

In the submucous plexus, the proportion of nNOS-immunoreactive neurons was doubled in the ileum and tripled in the colon, but not in the duodenum of diabetic rats, while the total neuronal number remained unchanged, suggesting neurochemical adaptation of submucous neurons [35]. These results emphasize that the diabetic state affects the two enteric plexuses differentially.

In addition, in the microenvironment of enteric neurons, the number of eNOS-labelling gold particles was increased in the capillary endothelium of different gut segments [36], suggesting that the microvessels supplying to the myenteric ganglia are targets of diabetic damage in a regional manner and may contribute to developing neuropathy in diabetes.

Sex dependency on the diabetic nitrergic dysfunction was also observed [37]; females seem to have greater vulnerability to diabetes-related gastric impairments than males [38]. A decreased level of tetrahydrobiopterin, a major cofactor for NO synthesis, contributes to delayed gastric emptying, reduced pyloric nitrergic relaxation and nNOS-α protein expression in female diabetic rats [39].

NOS-containing neurons go through a two-phase degeneration process. The decrease in axonal nNOS expression as the hallmark of the first phase is reversible, however, causing irreversible changes, as apoptotic loss of nitrergic neurons occurs in the second one [26,40]. In addition, progressive accumulation of advanced glycation end products (AGEs) during diabetes seems to enhance this apoptotic process [41]. AGEs significantly reduce the expression of nNOS and NO release in myenteric neurons via their receptor [42]. Prevention of AGE formation by certain drugs precedes the decrease of nNOS in diabetic rats [43] and may also help to protect against nitrergic nerve dysfunction. Furthermore, the endogenous antioxidant defense of the gut can also be protective for nitrergic myenteric neurons. An extensive increase in the ratio of nNOS-immunoreactive neurons colocalizing with heme oxygenases was revealed in the ileum and colon of diabetic rats, though the nitrergic neuronal number decreased [44], suggesting that those NOS neurons which do not colocalize with heme oxygenases are the most damaged by diabetes [44,45].

### 3.2. Chronic Alcohol Consumption

The impact of chronic ethanol intake on NOS-immunoreactive myenteric neurons in various gut regions has been investigated in our laboratory for more than a decade. The number of nNOS-immunoreactive neurons was significantly reduced in all investigated intestinal regions after 8 weeks of ethanol consumption [34]. Since the total neuronal number in the myenteric ganglia remained unchanged, the alteration in the number of nNOS-immunoreactive neurons means that ethanol treatment specifically affected nNOS production (Table 1), damaging NO pathways in the ENS and disturbing gastrointestinal motility [46]. A decreased number of nNOS-immunoreactive neurons was also described in the murine jejunum after alcohol intake, but because it is accompanied by an increased proportion of iNOS-immunoreactive neurons [47], the number of NO-producing neurons remained constant. This data proposed a compensatory mechanism to restore the NOS balance. However, using bio-imaging of individual myenteric neurons, the basal NO synthesis was significantly enhanced after chronic alcohol consumption. Bio-imaging recordings also strengthened the elevated NO synthesis in the enteric glial cells, smooth muscle cells and endothelial cells, indicating a general increase in NO production in the intestinal wall [23]. In cultures, NO facilitates the protective effects of neuronal growth factors and helps with developing neurons to resist alcohol toxicity by activating the NO-cGMP-PKG-NFkappaB signalling pathway [48,49].

Similarly, electron microscopic study of the distribution of different NOS enzymes after ethanol treatment revealed opposite changes in the quantity of nNOS- and eNOS-labelling gold particles not only in the myenteric ganglia, but also in their close microenvironment [18]. While the number of nNOS labels has fallen by more than half, eNOS labels almost doubled in the duodenal ganglia, suggesting possible functional plasticity between NOS isoforms that might help to maintain the optimum NO level during chronic alcohol consumption. Moreover, gut segment-specific rearrangement of NOS isoforms in the various subcellular compartments was detected in rats after ethanol treatment [18].

### 3.3. Intestinal Inflammation

Although impairments of enteric neurons in different inflammatory diseases have been widely studied [50,51,52,53,54,55], there are numerous open questions and no consensus on how the ENS is affected in these inflammatory disorders. Intestinal inflammation affects not only the density of enteric neurons but also their function [56,57].

Extensive damage of myenteric neurons as a distinct early event was demonstrated in the inflamed colonic segment in different rat models despite the sustained inflammatory processes [50,58]. The rapid influx of activated immune cells after the onset of colitis accompanied elevated NO production from enhanced iNOS expression, which can be cytotoxic to enteric neurons. The immune-cell-mediated neuronal loss could be blocked by iNOS inhibitors [59]. Glial cells can also enhance iNOS activity during inflammation [60]. Brown et al. [61] observed the enhancement of glial NO levels and increased immunoreactivity of nitrated proteins (another marker of NO concentration) colocalized with glial fibrillar acidic protein 48 h after the induction of colitis. Elevated nitric oxide facilitates the release of glial adenosine triphosphate and assists with neuronal loss [61].

Among the different subpopulations of enteric neurons, the vasoactive intestinal polypeptide (VIP)-positive and nitrergic neurons display contrary responses to gut inflammation. In the inflamed colonic regions, the number of NOS neurons increased and the percentage of VIP neurons was unchanged in the myenteric plexus of paediatric patients with Crohn’s disease [52]. However, in the submucous ganglia, an increased number of VIP-immunoreactive neurons was observed, while the number of NOS neurons was too low for quantification [52]. The regulation of nNOS expression seems different in models of Crohn’s disease and ulcerative colitis. In 2,4,6-trinitrobenzenesulfonic acid-induced Crohn’s disease, decreased nNOS protein and mRNA level was measured, but nNOS levels were not altered in experimentally-induced ulcerative colitis [53]. The proportion of nNOS-immunoreactive myenteric neurons was decreased in a stress-induced rat model of irritable bowel syndrome with diarrhea [54] contributing to the motility dysfunction characteristic for this disease.

The activity of NOS and the generation of NO were elevated in colonic mucosal biopsies [62] as well as rectal NO levels were markedly enhanced in active ulcerative colitis and Crohn’s disease [63]. Moreover, an increased number of mucosal nitrergic nerve fibers was revealed in the duodenum, jejunum and descending colon of dogs with irritable bowel disease, and this increment was proportional to the state of the disease ranging from mild to severe [64].

### 3.4. Ischaemic Injuries

The deleterious effects of intestinal ischemia/reperfusion (I/R) injury on enteric neurons was also revealed in several studies. Alterations in neuronal structures or even neuronal loss were demonstrated in humans [65] and animal models [66,67] with I/R.

However, evidence showed that the NOS-containing enteric neuronal subpopulation is more susceptible to severe enteric neuropathies [5], like I/R, suggesting that NO from nitrergic neurons contributes to neuronal damage. However, NO also has a neuroprotective effect: I/R resulted in more serious damage in nNOS knock out mice, than in their wild-type counterparts [68].

Swelling and distortion of nitrergic neuronal dendrites and increased relative cell profile area of NOS neurons was observed following I/R in the ischaemic intestinal region [69,70,71]. Moreover, protein nitrosylation and translocation of Hu protein (enteric neuronal marker) from the cytoplasm to the nuclei was also demonstrated shortly after the onset of I/R [72].

## 4. Conclusions

Several studies have suggested that nitrergic myenteric neurons are especially susceptible to the development of neuropathy in diseases of the digestive tract [5,27,28,34]. Although more than 20 years have passed since the biological properties of NO were discovered [73,74], many questions remain unanswered concerning the role of NO in neurons both in the gut and the brain [75]. To answer these questions, future studies will be required to investigate not only the ENS, but the microenvironment of the enteric ganglia, the microcirculation and neuro-immune interactions of the gut, the bidirectional communication in the gut–brain axis, and even the interactions among the microbiota, the GI tract and the brain in health and disease.

## Figures and Tables

**Figure 1 ijms-20-02003-f001:**
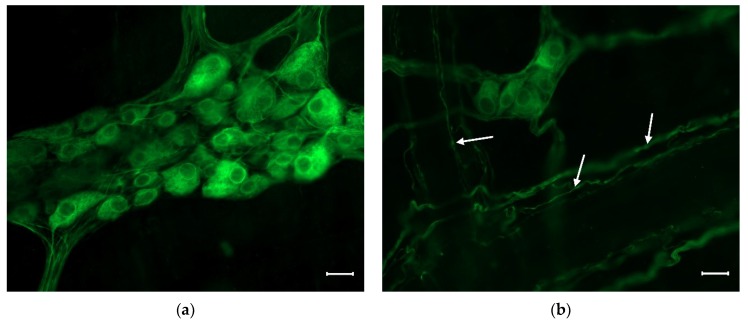
Representative fluorescent micrograph of a myenteric (**a**) and a submucous (**b**) ganglion from the duodenum of a control rat after peripherin immunolabelling. The myenteric ganglion (**a**) is bigger and contains more neurons than the submucous (**b**) one. Arrows show the immunolabelled neuronal processes in the submucous plexus. Scale bars: 20 µm.

**Figure 2 ijms-20-02003-f002:**
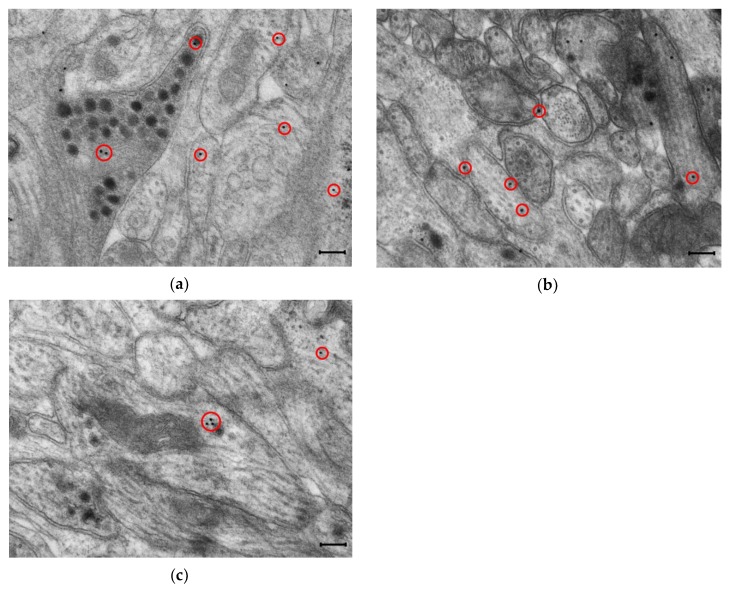
Representative transmission electron micrographs of myenteric ganglia from the duodenum of a control rat after post-embedding immunohistochemistry, using a nNOS (**a**), eNOS (**b**) and iNOS (**c**)-specific antibody. Circles show the 18 nm gold particles conjugated to different NOS isoforms. Scale bars: 200 nm.

**Figure 3 ijms-20-02003-f003:**
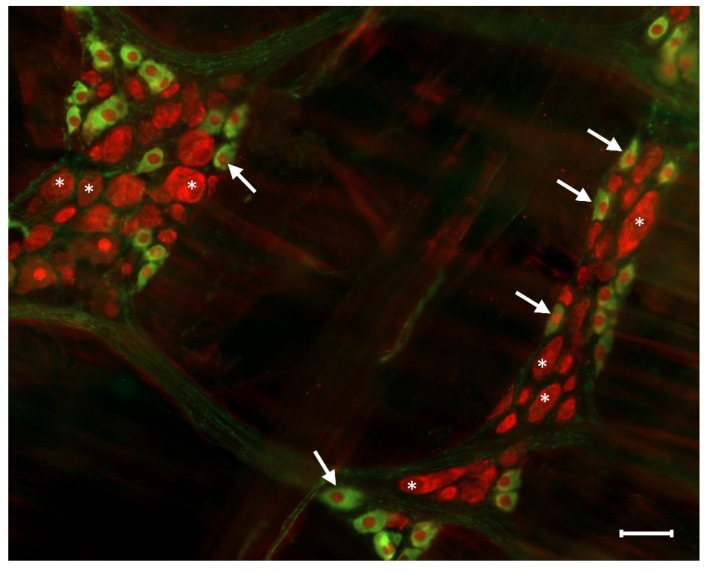
Representative fluorescent micrograph of myenteric neurons from the duodenum of a control rat after nNOS (green)-HuCD (red) double labelling. Stars indicate neurons that are labelled for HuCD only, arrows point to neurons that are double-labelled for both nNOS and HuCD. Scale bar: 50 µm.

**Table 1 ijms-20-02003-t001:** Alterations in the number of nitrergic and total myenteric neurons and their consequences in different intestinal regions of rats with type 1 diabetes or chronic alcohol consumption. The number of total and nitrergic neurons varied differently dependent on the gut segment and also the type of disease. The number of nitrergic myenteric neurons decreased in all investigated gut segments of diabetic rats and also ethanol-treated animals. In diabetic rats, with the exception of the duodenum, the total neuronal number was also decreased in the other intestinal regions; suggesting enteric neuronal loss in these segments and neurochemical modification in the duodenum. In ethanol-treated rats, the total number of myenteric neurons was not affected in the different gut segments proposing modification of nNOS production in neurons. Summarized from Izbéki et al. [27] and Krecsmarik et al. [34].

Animal Models	Number of Myenteric Neurons	Duodenum	Jejunum	Ileum	Colon
**Type 1 Diabetes [27]**	Nitrergic	↓	↓	↓	↓
	Total	Ø	↓	↓	↓
	Pathological alteration in nitrergic myenteric neurons	neurochemical modification	neuronal loss	neuronal loss	neuronal loss
**Chronic Alcohol Consumption [34]**	Nitrergic	↓	↓	↓	↓
	Total	Ø	Ø	Ø	Ø
	Pathological alteration in nitrergic myenteric neurons	neurochemical modification	neurochemical modification	neurochemical modification	neurochemical modification

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
