# Peer review of "Nitrergic Enteric Neurons in Health and Disease—Focus on Animal Models"

_ijms, 2019, doi:10.3390/ijms20082003_

Round 1
Reviewer 1 Report
The authors mini-reviewed the nitrergic neurons in the enteric nervous system. Overall, it is a neat, short and concise review which may benefit if they take the following into consideration:
1) It is not correct to state "NO is synthesized by nNOS, endothelial NOS (eNOS) and inducible NOS (iNOS) in the enteric neurons". Presence of mRNA of eNOS in enteric neurons does not mean that eNOS is expressed as an active protein. iNOS is expressed only when there is inflammation.
2) Some sections only give some facts without deliberating what this may mean; for example "The nitrergic neurons account for only a few 74 percent of all neurons in the submucous plexus [22], while the proportion of the nitrergic myenteric 75 neurons (Figure 3) is massive, and it is variable in different gut segments (23-52%, [13, 23])." What does this mean physiologically? can the authors speculate? Another example: "In addition, in the microenvironment of enteric neurons, the number of eNOS-labelling gold particles was increased in the capillary endothelium of different gut segments [33]."
3) In the section which starts with "The first evidence that the nitrergic neurons located in different gut segments display strictly region-specific responsiveness to the diabetic..." is a bit too much self-promotion; can it be shortened?
4) Page 5; two phases of nitrergic degeneration: the authors should mention that the shift from first phase to the second phase indicates irreversibility and happens if the diabetes is uncontrolled.
5) Spell out HO in first use.
6) Please avoid using the words "diabetic" or "diabetics". It should be "diabetic persons/rats".
7) Instead of quoting Nobel prize as a milestone, please quote the original works by Moncada et al and Ignarro et al who simultaneously discovered biological properties of NO.
Author Response
Answer to Reviewer 1 (in the order of paragraphs)
We would like to thank you for your valuable comments and suggestions; on the basis of them the manuscript was revised (corrections were highlighted by red colour in the revised version of the manuscript).
1) It is not correct to state "NO is synthesized by nNOS, endothelial NOS (eNOS) and inducible NOS (iNOS) in the enteric neurons". Presence of mRNA of eNOS in enteric neurons does not mean that eNOS is expressed as an active protein. iNOS is expressed only when there is inflammation.
Thank you very much your suggestion. In the revised manuscript this paragraph was changed (row 65-74).
2) Some sections only give some facts without deliberating what this may mean; for example "The nitrergic neurons account for only a few percent of all neurons in the submucous plexus [22], while the proportion of the nitrergic myenteric neurons (Figure 3) is massive, and it is variable in different gut segments (23-52%, [13, 23])." What does this mean physiologically? can the authors speculate?
The nitrergic neurons account for only a few percent of all neurons in the submucous plexus, while the proportion of the nitrergic myenteric neurons is significant. This massive proportion of the nitrergic myenteric neurons can be explained by the function of this neuron population. Most of the nitrergic myenteric neurons are inhibitory interneurons or inhibitory motoneurons innervating the muscle layers of the alimentary tract (row 81-87).
Alterations of eNOS expression in intestinal microvessels supplying to myenteric ganglia suggested that they are also targets of diabetic impairments and may contribute to developing neuropathy in diabetes. This sentence was expanded in 3.1. (row 116-119).
3) The study which details the region-specific responsiveness of nitrergic myenteric neurons in type 1 diabetes is one of the first results in our diabetes research. However, we did not want to be directly self-promoting. We agree with your opinion, therefore, we corrected this sentence and focused only on the paper results (row 96-98).
4) In the section about two phases of nitrergic degeneration, the irreversibility of changes in the second phase is indicated (row 124-126).
5) Heme oxgenases occur only twice in the text, therefore in the revised version we did not abbreviate it (row 133, 135).
6) Considering your request, we changed the “diabetics” word to “diabetic rats”.
7) Instead of quoting Nobel prize as a milestone, please quote the original works by Moncada et al and Ignarro et al who simultaneously discovered biological properties of NO.
Thank you very much your suggestion, it is better to cite both two significant papers (row 222-225).
We would like to thank you again for your precious remarks and hope that our answers and the revision of the manuscript will be accepted.
Sincerely yours,
Nikolett Bódi, Ph.D.
Reviewer 2 Report
Bódi et al. review the role of nitrergic enteric neurons in the gut with a major focus on the various causes and consequences for NO-neuron dysfunction/loss in gut neuropathies. It is a well-written, concise and up-to date review, that only show some minor overlap with the previous Rivera et al. review on the same topic (Neurogastroenterol Motil; 2011 23, 980-988). Moreover, the pictures are in general beautiful and informative.
For the reader to appreciate the great number of gut disorders where NOS1 neurons are dysregulated/lost, it could be helpful to mention also those not dealt with in this review. You could state in row 33 or row 83 for instance: “We will describe the role of NO and effect on NOS1-neurons in four conditions. Additional NOS1-neuron dysregulation in esophageal achalasia, hypertrophic pyloric stenosis and Chaga´s disease has been reviewed elsewhere (Rivera et al., …..).”
Some sections would benefit from adjustments as suggested below:
Fig 1b)
Stars are indicated to show capillaries around the submucosal plexi. How did you determine the presence of capillaries?
Row 56:
From this point of view, the ENS is similar (to) the central nervous system, since….
Comment: word missing.
Row 62-63:
Of the three NOS isoforms, nNOS constitutes the predominant source of NO in enteric neurons.
Comment: As NOS is the enzyme necessary for NO production, the sentence could be for example: “Of the three NOS isoforms, nNOS1 is the predominant form that produce NO in enteric neurons.”
Row 89-92
In diabetic rats, the nitrergic neuronal number was decreased to varying degrees in all investigated intestinal regions. In the jejunum, ileum and colon, the total number of myenteric neurons was also decreased assuming diabetes-related cell loss in these segments. However, in the duodenum of diabetics, the decreased number of nitrergic neurons were not accompanied by total neuronal loss, presuming neurochemical modification of neurons here [25].
Comment: This section is somewhat confusing. Second and first sentence are similar. Consider to remove first sentence and rephrase the remaining. For example: “In the jejunum, ileum and colon of diabetic rats, the total number of myenteric neurons was decreased. However, in the duodenum of diabetics, the decreased number of nitrergic neurons was greater than the total cell loss, suggesting region-specific neurochemical modifications.”
Row 93-96
Other results confirm the decrease of nitrergic myenteric neurons of the antrum and jejunum in spontaneously diabetic Bio-breeding rats [26], in the ileum of diabetic dogs [28], loss of nNOS neurons in the human appendix [29] and reduced gastric nNOS protein and mRNA expression [27, 30] in mice involving purinergic P2X7 receptors in diabetes-related damage of nitrergic neurons [30].
Comment: The sentence needs some restructuring. For example: “Several other studies confirm diabetes-associated decreases of nitrergic myenteric neurons. nNOS-neurons are reduced in the antrum and jejunum in diabetic Bio-breeding rats, in the ileum of diabetic dogs, in the appendix and stomach of diabetic humans, and in the stomach of the P2RX7-mediated diabetic mouse model.”
Row 112-113
AGEs significantly reduce the expression of nNOS and NO release in myenteric neurons via their receptor, receptor for AGEs.
Comment: unintentional repetition?
Row 136-137:
NO promotes the defense of neuronal growth factors and redound neuronal survival against alcohol toxicity via specific signaling pathways.
Comment: This sentence is unclear. Please rephrase.
Row 139
Similarly, electron microscopic study involved the distribution of different NOS enzymes after ethanol treatment reveal…
Comment: “involved” should be changed to “of”
Row 49-54
Impairments of enteric neurons in different inflammatory diseases have been thoroughly studied.
Comment: Please milden this statement. There is no consensus on how the ENS is affected in IBDs. Likely, detailed analysis with scRNA-seq or refined marker expression will determine the effect on ENS better. So far, studies have relyed on single neurotransmitters and lacked a large-scale systematic approach (enough patients, from several regions and states of the disease).
Row 182-183
Alterations of neuronal numbers or even neuronal loss were demonstrated in humans [64] and animal models [65, 66] with I/R.
Comment: Perhaps you meant to say “ Alterations in neural structures and even neuronal loss were demonstrated….”
Row:
However NO also has a neuroprotective effect: I/R resulted in more serious damage in nNOS knock out mice, than in their wild-type counterparts [68].
Comment: For consideration, knocking out nNOS has consequences on the whole gut; the worsen phenotype might be indirect and not directly linked to the absence of nNOS.
Author Response
Answer to Reviewer 2 (in the order of paragraphs)
First of all, we would like to thank you for your valuable comments and suggestions; on the basis of them the manuscript was revised (corrections were highlighted by red colour in the revised version of the manuscript).
We agree with your remark that the nNOS neurons are damaged in more gastrointestinal disorders than we described in the present review. Therefore, according to your suggestion, other intestinal diseases with nNOS involvement are listed with reference in row 33-37.
Figure 1.b was made after single labelling with anti-peripherin antibody, we could not determine immunohistochemically the presence of capillaries, therefore the star and the mention of the capillary was deleted from figure legend in the revised version of the manuscript (row 46).
Considering your other adjustments:
§ The missing word (to) was inserted in row 61.
§ We corrected the sentence in row 71-73 that nNOS is the predominant form of NO production in enteric neurons.
§ To comply with your request, the section between rows 98-104 was reworded and the first sentence was deleted.
§ The sentence between rows 104-109 was restructured.
§ In row 129, the redundant repetition of AGE receptor was deleted.
§ In row 152-155, the sentence has been reworded and clarified.
§ The word ‘involved’ changed to ‘of’ in row 156.
§ According to your request, the first sentence of this section (row 177-179) emphasizes the numerous open questions about how the enteric neurons are affected by intestinal inflammation.
§ The sentence was corrected in row 209-210.
§ Thank you very much your completion for rows 213-214.
We would like to thank you again for your precious remarks and hope that our answers and the revision of the manuscript will be accepted.
Sincerely yours,
Nikolett Bódi, Ph.D.
Reviewer 3 Report
The authors give an overview on some aspects of NOS in the enteric nervous system. The following comments should be considered and processed before publication.
During the last three decades, a huge amount of literature has been published both on NOS and the ENS. The title „Nitrergic neurons in health and disease“ promises a lot, including some more reference to the human ENS as well as NOS in the human ENS. By title (and apart from some reviews cited), only 6 of the 73 literature sources cited refer to studies presenting results obtained from human tissues (e.g., rat: 20, mouse: 9, other: 8). In my opinion, there are two possibilities. Either, the manuscript must be drastically extended and rewritten. Or, the title should indicate that some aspects are emphasized, e.g. „Nitrergic neurons in health and disease – focus on …“ (maybe a „minireview“?). In both cases, any kind of schematic drawing or a table to be included would fit well.
Author Response
Answer to Reviewer 3 (in the order of paragraphs)
We would like to thank you for your valuable comments and suggestions; on the basis of them the manuscript was revised (corrections were highlighted by red colour in the revised version of the manuscript).
We agree with your opinion that the manuscript focuses on nitrergic enteric neurons and NOS in the different pathological animal models, and only a few cited studies refer to the humans. Unfortunately, the human results in the topic of NOS in enteric neurons are poorly, and on the other hand, our research group apply also animal models to investigate different gastrointestinal diseases. Therefore, to comply with your request, we decided to change the title of the manuscript emphasizing the focus on the pathological animal models.
Considering your suggestion a new table (Table 1, see in the attached PDF file) was inserted in the revised manuscript (row 164-174). This table summarizes the changes in the number of nitrergic and total myenteric neurons, as well as it concludes the meaning of these alterations in different intestinal regions and animal models.
We would like to thank you again for your precious remarks and hope that our answers and the revision of the manuscript will be accepted.
Sincerely yours,
Nikolett Bódi, Ph.D.

Round 2
Reviewer 3 Report
The authors have changed the title. However, they should delete "pathological" from the title: "health and disease" in the first part as well as "animal model" in the second seems sufficient.
Author Response
Answer to Reviewer 3
We would like to thank you for your suggestion.
According to your request, the “pathological“ word was deleted from the title (correction was highlighted by green colour in the revised version of the manuscript).
Sincerely yours,
Nikolett Bódi, Ph.D.